# Modeling Conceptual Understanding in Image Reference Games

**Rodolfo Corona**[*][†]
UC Berkeley
rcorona@berkeley.edu

**Stephan Alaniz**[*][†]
Max Planck Institute for Informatics
salaniz@mpi-inf.mpg.de

**Zeynep Akata**[†]
University of Tübingen
zeynep.akata@uni-tuebingen.de

## Abstract

An agent who interacts with a wide population of other agents needs to be aware that there may be variations in their understanding of the world. Furthermore, the machinery which they use to perceive may be inherently different, as is the case between humans and machines. In this work, we present both an image reference game between a speaker and a population of listeners where reasoning about the concepts other agents can comprehend is necessary and a model formulation with this capability. We focus on reasoning about the conceptual understanding of others, as well as adapting to novel gameplay partners and dealing with differences in perceptual machinery. Our experiments on three benchmark image/attribute datasets suggest that our learner indeed encodes information directly pertaining to the understanding of other agents, and that leveraging this information is crucial for maximizing gameplay performance.

## 1 Introduction

For a machine learning system to gain user trust, either its reasoning should to be transparent [Rudin, 2019, Freitas, 2014, Lakkaraju et al., 2016, Letham et al., 2015], or it should be capable of justifying its decisions in human-interpretable ways [Gilpin et al., 2018, Hendricks et al., 2016, Huk Park et al., 2018, Wu and Mooney, 2019]. If a system is to interact with and justify its decisions to a large population of users, it needs to be cognizant of the variance users may have in their conceptual understanding over task-related concepts, i.e., an explanation could make sense to some users and not to others. Although there has been work studying what affects users' ability to understand the decisions of machine learning models [Chandrasekaran et al., 2017], to the best of our knowledge existing work in explainable AI (XAI) does not explicitly reason about user understanding when generating explanations for model decisions.

As an additional complication, variations in understanding can only be inferred from observed behavior, as the system typically has no access to the internal state of its users. Further, usually not only the understanding among the population of users vary, but also how the system and its users perceive information about the world significantly differs, as is the case between human eyes and digital cameras artificial agents use for perception.

In this work, we focus on the ability of a machine learning system, i.e. an agent, to form a mental model of the task-related, conceptual understanding other communication partners have over their environment. Particularly, we are interested in an agent that can form an internal, human-interpretable

---

[*]Equal contribution  [†] Majority of work done at the University of Amsterdam

representation of other agents that encodes information about how well they would understand different descriptions presented to them. Further, we would like our agent to be capable of forming this representation quickly for novel agents that it encounters. Similar to Rabinowitz et al. [2018], we wish to generate a representation of other agents solely from observed behavior. Rather than implicitly encoding information about the agents' policies, we explicitly encourage our learned representation to encode information about their understanding of task-related concepts. We accomplish this through a value function over concepts conditioned on observed agent behavior, yielding a human-interpretable representation of other agents' understanding.

As a testbed, we formulate an image reference game played in sequences between pairs of agents. Here, agents are sampled from a population which has variations in how well they understand different visual attributes, necessitating a mental model over other agents' understanding of those visual attributes in order to improve the overall game performance. For example, an agent might understand color attributes poorly, leading it to have trouble differentiating between images when they are described in terms of color. We present ablation experiments evaluating the effectiveness of learned representations, and build simple models for the task showing that actively probing agents' understanding leads to faster adaptation to novel agents. Further, we find that such a model can form clusters of agents that have similar conceptual understanding.

With this work, we hope to motivate further inquiry into models of conceptual understanding. Our exemplar task, i.e. image reference game, based on real-world image data allows us to explore and observe the utility of agents who are able to adapt to others' understanding of the world.

## 2 Related Work

**Modeling Other Agents.** Inspired by Rabinowitz et al. [2018], we would like to model another agent solely from observed behavior, focusing on forming representations which encode information about their understanding of task-related concepts.

Recent works have also employed a similar idea to other multi-agent settings. In [Shu and Tian, 2019], an agent learns the abilities and preferences of other agents for completing a set of tasks, however, in their work they assume that the identities of the agents the learner interacts with are given and that their representation is learned over a large number of interactions. In contrast, we are interested in a learner that can quickly adapt to agents without having prior knowledge of who they are. The model presented by Shu et al. [2018] learns how to query the behavior of another agent in order to understand its policy. However, in their work only the environmental conditions vary, with the agent being modeled remaining the same. Here, we vary both agent and environment. There also exists a body of work on computational models of theory of mind [Butterfield et al., 2009, Warnier et al., 2012], particularly employing Bayesian methods [Baker et al., 2011, Nakahashi et al., 2016, Baker et al., 2017], although they use discrete state spaces rather than continuous ones.

**Meta Learning.** In meta-learning [Schmidhuber, 1987, Bengio et al., 1992, Finn et al., 2017], an agent is tasked with learning how to solve a family of tasks such that it can quickly adapt to new ones. In our work, we are interested in an agent that can learn to quickly adapt to the conceptual understanding of novel gameplay partners whose understanding is correlated to other agents from the population (e.g. such as learning to identify when someone is color-blind).

**Emergent Language.** There have been a number of works presenting multi-agent systems where agents must collaboratively converge on a communication protocol for specifying goals to each other [Choi et al., 2018, Evtimova et al., 2018, Foerster et al., 2016, Havrylov and Titov, 2017, Jorge et al., 2016, Lazaridou et al., 2017, 2018, Das et al., 2017, Kottur et al., 2017]. Whereas in these works the main focus is to learn an effective communication protocol and to analyze its properties, here we are interested in modeling other agents' understanding of the environment. We therefore assume a communication protocol is given so that we test agent modeling in isolation. Further, many of these works assume that gradients are passed between agents. Here, we assume a discrete bottleneck in that agents only have access to observations of each other's behavior. Although some domains have a population of agents [Mordatch and Abbeel, 2018, Cogswell et al., 2019], the tasks do not use real images and all agents either share a single policy or have equal capacity to understand task-related concepts. We believe that incorporating an emergent communication component to our domain would be an exciting avenue for future work.

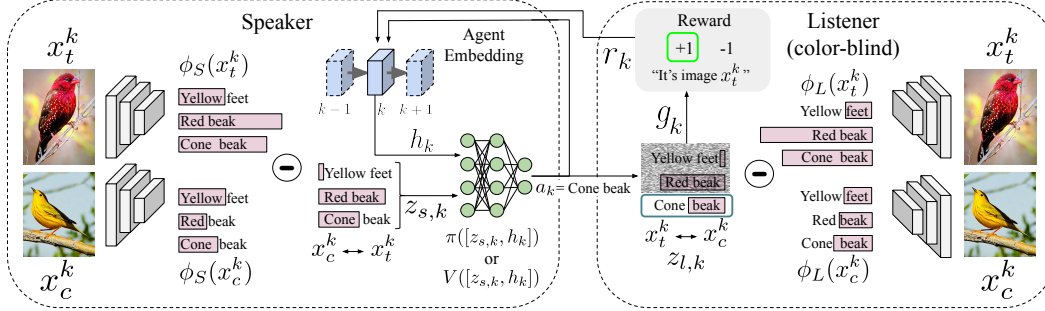

Figure 1: Our image reference game with varied agent population. In a given episode $k$, the speaker and listener encode the image pair $(x_t^k, x_c^k)$ using their perceptual modules $\phi_S, \phi_L$. The speaker selects a target image $x_t^k$ and an attribute $a_k$ to describe it using parameterized functions $\pi_S$ and $V$ conditioned on the image representations and agent embedding $h_{k-1}$. Given $a_k$, the listener guesses the target image. Finally, the speaker incorporates information about the listener into embedding $h_k$ given the reward $r_k$ received for using $a_k$ in that game.

# 3 Image Reference Game with Varied Agent Population

In a multi-agent communication setting, it is generally best to send a message which maximizes the amount of task-related information, such as describing an image by appealing to its most discriminative attributes. However, the recipient of the message may not be familiar enough with certain attributes, meaning that some messages are not useful to them despite being maximally informative. In line with this motivation, we formulate an image reference game where agents must describe images to each other using visual attributes.

**Task definition.** In our visual reference game (see Figure 1) we have a single learner, referred to as the *speaker*, who must learn to play sequences of episodes in an image reference game with gameplay partners, referred to as *listeners*, that are randomly sampled from a population of agents. Both the speaker and the listeners are given a pair of images in each episode $k$, and the speaker selects an image $x_t^k$ to serve as the target, with the second image $x_c^k$ serving as confounder. The speaker must then generate a description, in the form of an image attribute $a_k \in A$, which the listeners use to compare the two images before guessing the target's identity.

As listeners are effectively black-boxes to the speaker, it can be difficult to disentangle potential sources of error when they behave unexpectedly. Namely, when a listener guesses incorrectly, it is difficult to tell whether the mistake was due to a lack in its understanding of 1) the game, 2) the language used to communicate, or 3) the attribute (i.e. concept) used to describe the image. In this work, we focus on the third option (conceptual understanding), and isolate this problem from the other two by assuming that the speaker can communicate attribute identities noiselessly, and that listeners are all rational game players sharing a static gameplay policy.

**Perceptual Module.** An agent's perceptual module $\phi$ encodes images into a list of shared concepts weighted by their relevance to the image. Specifically, the perceptual module first extracts image features using a CNN. The image features are further processed with a function $f$ that predicts attribute-level features $\phi(x) = f(\text{CNN}(x))$, where $\phi(x) \in [0, 1]^{|A|}$, and $|A|$ is the number of visual attribute labels in an attribute-based image classification dataset.

Every element in $\phi(x)$ represents a separate attribute, such as "black wing", giving us a disentangled representation. The speaker and listener policies reason about images in the attribute space $A$; we are interested in disentangled representations because they will allow for the speaker's mental model of listeners' understanding to be human interpretable. In our setting, the speaker is given a separate module $\phi_S$, while all listeners share a single module $\phi_L$.

## 3.1 Modeling Listener Populations

If a listener has a good understanding of an attribute, we would expect that it would be able to accurately identify fine-grained differences in that attribute between a pair of images. For example,

someone with a poor understanding of the attribute "red" may not be able to distinguish between the red in a tomato and the red in a cherry, although they might be capable of distinguishing between the redness of a fire truck and that of water. Following this intuition, we generate a population of listeners $L = \{(\delta_l, p_l)\}$, where each listener $l \in L$ is defined by a vector of thresholds $\delta_l \in [0, 1]^{|A|}$ and a vector of probabilities $p_l \in [0, 1]^{|A|}$.

Given an image and attribute feature pair $\left(\phi_L(x_t^k), \phi_L(x_c^k)\right)$, the listener $l$ first computes the difference between the attribute features $\phi_L$ of image $x_t$ and $x_c$ for attribute $a$:

$$z_l^a = \phi_L^a(x_t^k) - \phi_L^a(x_c^k). \tag{1}$$

Using its attribute-specific threshold $\delta_l^a$, if $|z_l^a| < \delta_l^a$, then the listener does not understand the concept well enough and will choose the identity of the target image uniformly at random. Conversely, if $|z_l^a| \geq \delta_l^a$, then the listener will guess rationally with probability $p_l^a$ and randomly with probability $(1 - p_l^a)$. Here, a rational guess $g = \arg\max_{x \in \{x_t^k, x_c^k\}} \phi_L^a(x)$ means choosing the image which maximizes the value of the attribute $a$.

To simplify the setup, we specify a total of two different levels of understanding. An agent can either understand an attribute, i.e. $u = (\delta, p)$, or not understand an attribute, i.e. $\bar{u} = (\bar{\delta}, \bar{p})$. For $u$, $\delta$ is small and $p$ is set to 1, respectively meaning that attributes are easily understood and the agent always plays rationally on understood attributes. Conversely, $\bar{u}$ specifies a high value for $\bar{\delta}$ and $\bar{p}$ is lower than 1, such that an attribute that is not understood rarely leads to rational gameplay.

To form a diverse population of listeners, we create a set of clusters $C$ where each cluster is defined by the likelihood of assigning either $u$ or $\bar{u}$ to each individual attribute. Thus, listeners sampled from the same cluster will have correlated sets of understood and misunderstood attributes, while remaining diverse.

## 3.2   Modeling the Speaker

In a given sequence, the speaker plays $N$ practice episodes, each consisting of a single time-step, where the purpose is to explore and learn as much about the understanding of the listener as possible, purely from observed behavior. During the $k$'th game in a sequence with a given listener, the speaker first encodes the image pair with $\phi_S$. From the previous $k - 1$ games, the speaker also has access to an agent embedding $h_{k-1}$, which encodes information about the listener. The speaker uses an attribute selection policy to select an attribute $a_k$ for describing the target image. After the listener guesses, the reward $r_k$ from the game is used to update the agent embedding into $h_k$. After the practice episodes, $M$ evaluation episodes are used to evaluate what the speaker has learned.

**Agent Embedding Module.** To form a mental model of the listener in a given sequence of episodes, the speaker makes use of an agent embedding module. This module takes the form of an LSTM [Hochreiter and Schmidhuber, 1997] which incorporates information about the listener after every episode, with the LSTM's hidden state serving as the agent embedding. Specifically, after selecting an attribute $a_k$ and receiving a reward $r_k \in \{-1, 1\}$, a one-hot vector $o_k$ is generated, where the index of the non-zero entry is $a_k$ and its value is $r_k$. The agent embedding $h_k = \text{LSTM}(h_{k-1}, o_k)$ is then updated by providing $o_k$ to the LSTM.

**Attribute Selection Policies.** The speaker has access to two parameterized functions, $V(s_k, a_k)$ and $\pi_S(s_k, a_k)$, represented by multi-layer perceptrons. The speaker uses these functions to select attributes during the $N$ practice and $M$ evaluation episodes, where $s_k = \left[\phi(x_t^k) - \phi(x_c^k); h_k\right]$ is a feature generated by concatenating the image-pair difference and agent embedding.

We estimate the value of using each attribute to describe the target image, i.e. $V(s_k, a_k) : \mathcal{R}^d \times \mathcal{A} \to \mathcal{R}$ using episodes from both the practice and evaluation phases optimizing the following loss:

$$\mathcal{L}_V = \frac{1}{N + M} \sum_{N+M} \text{MSE}(V(s_k, a_k), r_k) \tag{2}$$

As $V$ approximates the value of each attribute within the context of a listener's embedding and an image pair, it directly provides a human-interpretable representation of listeners' understanding. Therefore, every model presented uses it greedily to select attributes during evaluation games.

The purpose of practice episodes is to generate as informative an agent embedding as possible for $V$ to use during evaluation episodes. Therefore, speakers differ in how they select attributes during

practice episodes, probing listeners' understanding with different strategies. One strategy is to use an attribute selection policy $\pi_S$, trained with policy gradient [Sutton et al., 2000], which directly maps to probabilities over attributes. In the following, we describe different attribute selection strategies used during practice episodes.

**a. Epsilon Greedy Policy.** For this selection policy, we simply either randomly sample an attribute with probability $\epsilon$ or greedily choose the attribute $a_k = \arg\max_{a \in A} V(s_k, a)$ using $V$.

**b. Active Policy.** The active policy is trained using policy gradient:

$$\mathcal{L}_a = \frac{1}{N} \sum_N -R \log \pi_S(s_t, a_t) \text{ with } R = -\frac{1}{M} \sum_M \text{MSE}(V(s_k, a_k), r_k) \qquad (3)$$

where the reward ($R$) for the policy is a single scalar computed from the evaluation episode performance. This encourages the policy to maximize the correctness of the reward estimate function $V$ during evaluation episodes, requiring the formation of an informative agent embedding during the practice episodes. Note that when optimizing the active policy $\pi_S$, gradients are not allowed to flow through $V$.

## 4   Experiments

In the following, we first evaluate the effects of using different attribute selection strategies during practice episodes and then the quality of agent embeddings generated by each model. We use the AwA2 [Xian et al., 2018], SUN Attribute [Patterson et al., 2014], and CUB [Wah et al., 2011] datasets. Unless stated otherwise, the listener population consists of 25 clusters, each with 100 listeners. We use two variants of the perceptual module, ResNet-152 [He et al., 2016], fine-tuned for attribute-based classification with an ALE [Akata et al., 2013] head, and PNASNet-5 [Liu et al., 2018] with an attribute classifier head. Both ResNet and PNASNet-5 are pre-trained on ImageNet [Deng et al., 2009] and fine-tuned for the attribute-based image classification task. Note that unless stated otherwise, in each experiment both the speaker and listeners use the same perceptual module, i.e. $\phi_S = \phi_L$.

For all curves we plot the average over 3 random seeds, with error curves representing one standard deviation. We use the standard splits for CUB and SUN, but make our own split for AwA2 in order to have all classes represented in both train and test. The training splits are used for learning speaker parameters; we present performance on the test splits, using the same splits for each seed. We sample target and confounder images from the same dataset split. Listener clusters $C$ are shared across train and test but a novel population of listeners is sampled at test time[2].

### 4.1   Policy Comparison

We first compare the performance of the Epsilon Greedy and Active policies described in Section 3.2 against three baselines, the Random Agent, Reactive, and Random Sampling policies.

Among the baselines, the Random Agent policy simply always selects an attribute at random to describe images. The Reactive policy, at the beginning of each set of $N + M$ episodes, randomly selects an attribute. It continues using this attribute for each episode, only sampling a different attribute whenever it encounters a negative reward, keeping track over which attributes it has used. This policy is meant as a sanity check against a degenerate strategy of only using the LSTM to remember which attributes have worked and which have not, without incorporating useful information about the listener's conceptual understanding. Finally, the Random Sampling baseline selects random attributes during practice episodes, and then follows a greedy strategy over $V$ during evaluation episodes.

The performance of these policies on the test set is presented in Figure 2 which shows that the Epsilon Greedy and Active selection policies both outperform the Reactive baseline, suggesting that the agent embedding is encoding information about the conceptual understanding of listeners. After a large number of games, we would expect the performance of the Epsilon Greedy, Active and Random Sampling policy to be the same because at some point the speaker agent has learned about all the listener's understood and misunderstood attributes. By comparing against the Random Sampling policy, we can conclude that both the Epsilon Greedy and Active policies can learn more efficient

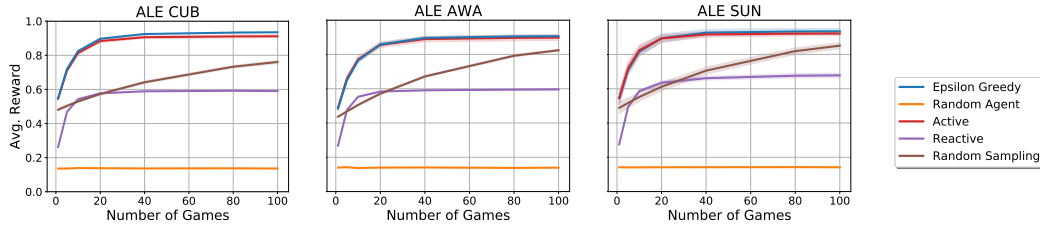

Figure 2: A comparison of average test set performance (Avg. Reward) for different attribute selection policies vs. the number of practice games. All agents learn from the listeners responses, i.e. using an embedding module, except for the random agent which always acts randomly. With an increasing number of games, the agent observes more responses providing information about the listener's conceptual understanding.

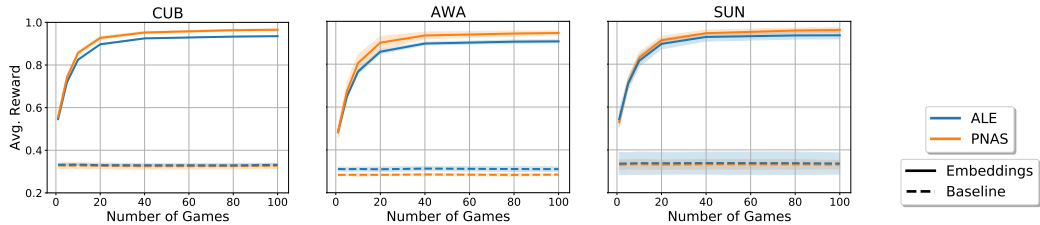

Figure 3: Ablation study on the importance of the agent embedding module. Average reward of the Epsilon Greedy policy on the test set as the number of practice episodes played increases. We evaluate the performance on two different perception modules (ALE, PNAS) with embedding module and without (baseline).

strategies that identify the misunderstood attributes within the first 20 games, at least five times faster than the Random Sampling policy. This corroborates the positive effect of encouraging policies to query information that helps the speaker form a mental model of the listener.

## 4.2 Evaluating Agent Embedding

Here we present an ablation study to investigate the benefit of using agent embeddings when playing the game, training an epsilon-greedy policy for each dataset until convergence with (Embeddings) and without (Baseline) agent embeddings.

Models without agent embeddings are given zero vectors, $h_k = 0$, instead of agent embeddings as input for the attribute selection policies. In these experiments, the speaker and listeners share the same perception module; we test performance for both the ALE and PNAS perceptual modules. Intuitively, a speaker will improve its performance over the game sequence if it encodes useful information about the listener, since it will help it avoid using attributes which the listener does not understand well.

In Figure 3, we show the average reward at different intervals of the game sequence. Using an agent embedding module significantly improves the performance of the speaker over time in all cases. Most importantly, performance improves as the number of games increases, showing that a speaker using an agent embedding module can quickly adapt to individual listeners from experience to avoid using misunderstood attributes and, thus, achieve a higher average reward.

## 4.3 Evaluating Cluster Quality

Although we have shown that agents with an agent embedding module achieve better performance, these results do not necessarily imply that speakers with memory develop an informative mental model over the conceptual understanding of the listeners. In order to test this, we perform an additional experiment on the trained speaker models. Specifically, we play roughly 50K sequences on the test set in order to generate a dataset of agent embeddings. We then perform K-Means clustering on these embeddings with $k = |C|$ (i.e. the number of listener clusters in the population) to obtain cluster assignments $C'$ and compare them to the ground-truth listener cluster assignments $C$.

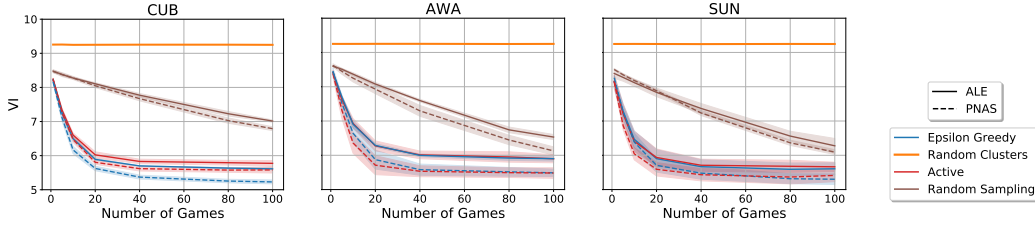

Figure 4: Variation of information ($VI$) of agent clusters $C'$ compared to ground-truth cluster assignments $C$. We present $VI$ for different policies as the number of practice games increases, lower is better. Cluster assignments $C'$ are obtained via K-Means ($k = |C|$) on agent embeddings from 50K test set sequences for each policy. Random Clusters (baseline) assigns each embedding to a random cluster. Each policy is evaluated using two different perception modules (ALE, PNAS).

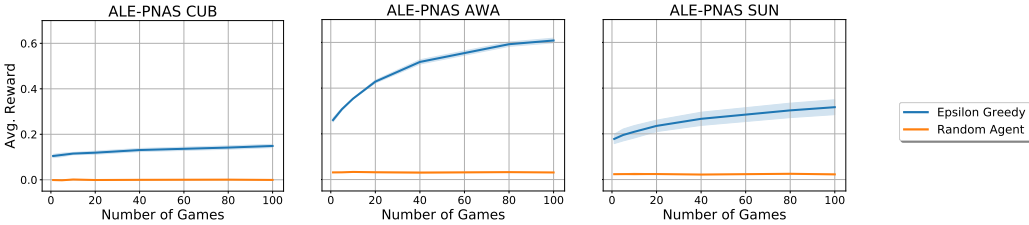

Figure 5: Average test set performance over different evaluation intervals for Epsilon Greedy and a random baseline. Here we test giving the speaker and listener population different perceptual modules (speaker uses ALE, listener uses PNAS).

To evaluate the cluster quality, we use the variation of information (VI) metric [Meilă, 2003]:

$$VI(C, C') = H(C) + H(C') - 2I(C, C') \tag{4}$$

Here, $C$ and $C'$ are two different clusters, $H$ is the entropy, and $I$ is the mutual information. Intuitively, the $VI$ measures how much information is lost or gained by switching from clustering $C$ to $C'$.

The more informative agent embeddings are about listeners' understanding, the greater the correlation will be between the inferred cluster and the ground-truth cluster. Figure 4 shows clustering performance for all parameterized policies as the number of practice games increases per sequence. We additionally compare this performance to a random cluster assignment baseline.

Firstly, we note that every policy outperforms the random assignment baseline. The Epsilon Greedy and Active policies experience nearly identical gameplay performance, suggesting that simply optimizing for reward yields similarly informative embeddings as more explicitly encouraging the policy to maximize the value function's accuracy. Finally, the Random Sampling baseline converges much more slowly, corroborating the idea that a more directed exploration of listeners' capabilities proves useful. Due to the significant improvement over random cluster assignments, we conclude that the speaker agent learns an embedding that clusters the listeners similar to the ground truth. This suggest that the agent not only learns from previous games, but it also forms a more general representation of listener groups with similar conceptual understandings.

## 4.4 Evaluating Different Perceptual Modules

If our speaker is to interact with a varied population of agents, it not only needs to be cognizant that those it interacts with could have varying levels of understanding; the population itself could have inherently different machinery for perceiving the world, as is the case between humans and machines.

Therefore, we repeat the experiment from section 4.1 with the Epsilon Greedy policy, and give the speaker and the listener population different perceptual modules. Specifically, in Figure 5, we show test performance when assigning ALE to the speaker and PNAS to the listeners comparing to a speaker which randomly selects attributes.

We observe a drastic change in performance, which suggests that the difficulty of the problem significantly increases when the speaker and listeners have fundamentally different perception.

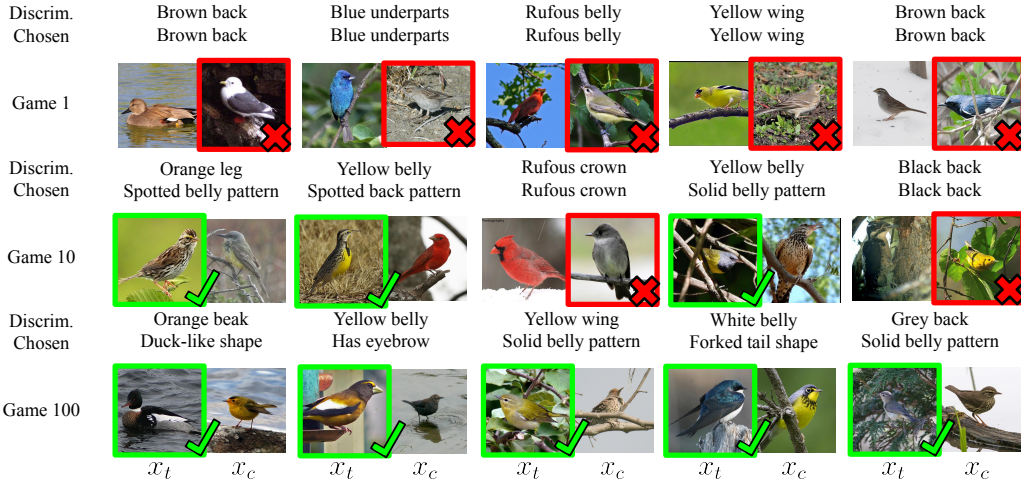

| | | | | | |
|---|---|---|---|---|---|
| Discrim.<br>Chosen | Brown back<br>Brown back | Blue underparts<br>Blue underparts | Rufous belly<br>Rufous belly | Yellow wing<br>Yellow wing | Brown back<br>Brown back |

Game 1

| | | | | | |
|---|---|---|---|---|---|
| Discrim.<br>Chosen | Orange leg<br>Spotted belly pattern | Yellow belly<br>Spotted back pattern | Rufous crown<br>Rufous crown | Yellow belly<br>Solid belly pattern | Black back<br>Black back |

Game 10

| | | | | | |
|---|---|---|---|---|---|
| Discrim.<br>Chosen | Orange beak<br>Duck-like shape | Yellow belly<br>Has eyebrow | Yellow wing<br>Solid belly pattern | White belly<br>Forked tail shape | Grey back<br>Solid belly pattern |

Game 100

$$x_t \quad x_c \qquad x_t \quad x_c \qquad x_t \quad x_c \qquad x_t \quad x_c \qquad x_t \quad x_c$$

Figure 6: Qualitative examples of the Epsilon Greedy agent on CUB interacting with a color-blind listener. Red (green) indicates an incorrect (correct) pick by the listener. We show examples where the speaker loses the first game due to selecting a discriminative color attribute. Even though color attributes are objectively more discriminative, in game 10 the speaker communicates color attributes less frequently. Finally, at convergence, i.e. game 100, the speaker prominently mentions shape-based attributes or non-color patterns.

Notice, however, that the performance of the Epsilon Greedy policy still significantly outperforms the random baseline. Further, particularly in the case of the Animals with Attribute dataset, the Epsilon Greedy speaker is still able to improve its performance as the number of episodes increases. This motivates further work in models that are capable not only of reasoning about conceptual understanding but also of adapting to fundamental differences in perception.

### 4.5 Qualitative Example

To provide an illustrative example of our reference game and the behavior of the agents, we train an Epsilon Greedy policy on the CUB dataset with 5 listener clusters, pertaining to the 5 attribute types found in the dataset (i.e. color, shape, size, pattern, and length). Each cluster in the listener population has a generally poor understanding of the attribute type it is assigned (e.g. the color cluster is color-blind). We visualize the center crop of the images as presented to both the speaker and the listener populations.

In Figure 6, we show sequences of games with color-blind listeners, where we can observe how the speaker adapts its strategy as it learns more about its gameplay partner – specifically, it adapts to using non-color attributes even in cases where color attributes would generally be most discriminative. In the first game, the speaker refers to objectively very discriminative color attributes such as brown back and rufous belly (columns 1 and 3). By game 10, the speaker already chooses color-invariant patterns over color attributes for some of the color-blind listeners, e.g. pointing out a spotted belly pattern over orange legs (column 1). After 100 games, we observe that the speaker almost always refers to non-color attributes, such as the duck-like shape or the presence of an eyebrow (column 1 and 2) because it leads to a higher average reward for color-blind listeners.

## 5 Conclusion

In this work, we presented a task in which modeling the understanding that other agents have over concepts is necessary in order to succeed. Further, we provide a formulation for an agent that is capable of modeling other agents' understanding and can represent it in a human-interpretable form. We believe that the ability to perform this kind of reasoning will allow XAI systems to tailor their explanations to the specific users with whom they interact. Learned agent embeddings can allow us to recover a clustering over other agents' conceptual understanding, which is a promising result to further tie this information into explanations. For example, by having explanations that are fitted to

each cluster, generated explanations would be more easily digestible by users of the system. Further, we show that naively modeling this type of reasoning is not sufficient for cases where the perceptual machinery of the learner and the population is fundamentally different.

**Acknowledgements** This work has received funding from the ERC under the Horizon 2020 program (grant agreement No. 853489), DFG-EXC-Nummer 2064/1-Projektnummer 390727645 and DARPA XAI program. R. Corona was supported in part by the Fulbright U.S. Student Program.

## Footnotes

[2]Code with full specifications for experiments may be found at: `https://github.com/rcorona/conceptual_img_ref`

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
