[Reviews · NeurIPS 2019]

Reviewer 1



— The proposed task, agent architecture, experiments, and analysis are all clearly described, and the writing overall is easy to follow (which is great!). — The proposed task is inspired by and is a simple adaptation of prior work in visual reference games. Each of the two agents (speaker and listener) gets to see a pair of images (target and confounder). The listener agent understands various visual attributes to varying levels of accuracy. During the training phase, the speaker agent gets to see observations of the listener's predictions / obtained rewards and must build a mental model for the listener agent. During the evaluation phase, the speaker must decide which visual attribute to emit in order to maximize the chances of the listener being able to pick the target image. — Experimental evaluation is technically sound. The comparisons with and without the agent embedding, different policies, and different perception modules (all across datasets) are all quite insightful, and validate the hypothesis that an agent that builds a mental model of its partner's abilities works better than an agent which does not.

Reviewer 2



Summary --- Consider a speaker agent and many listeners where listeners perceive differently (e.g., some know what cat furr looks like and others don't). This paper proposes an image reference game and develops a speaker that performs better at the reference game by modeling listener abilities. (motivation) AI agents should model the people they interact with because different people have different abilities. For example, one person might be able to visually classify many specific dog breeds wheras another person might not know anything about what dogs look like. (approach) Agents and Interaction: The paper sets up an image reference where both speaker and listener get to see the same pair of images and the listener needs to guess which image is the target (known to the speaker). The speaker utters image attributes which the listener uses to distinguish between the two images. Reference Game Flow: There are two stages of interaction analogous to meta-learning setups: practice and evaluation. In the practice stage the speaker gets utters lots of attributes to see which image they make the listener choose. In the evaluation stage a new set of images is used to measure how often the listener chooses the target. Perception: Each listener's perception is randomly corrupted so it cannot perceive some attributes very well, meaning the speaker shouldn't identify the target image using those attributes. Perception is shared between listener and speaker except for the noise added to listener perception. Speaker Implementation: Speakers judge a listener's ability by embedding the listener with an LSTM based on its performance at practice time, completing the meta-learning flavor of the approach. Speaker models are implemented either using an action-value function or a policy trained via policy gradients. (experiments) The evaluation measures how variations in speaker models affect listeners' abilitiy to predict the target image. 1. Both proposed speaker models perform quite well and drastically outperform some dumb baselines. 2. The proposed speaker model far outperforms a version with listener embeddings ablated, suggesting the proposed method usefully models listener expertise. 3. Listener embeddings learned by the speaker mimic the structure used to create the population of listeners. 4. If listeners and speaker learn different perception modules then performance dramatically decreases, though it still outperforms simple baselines. Strengths --- In addition to the contributions above: * This work is well positioned to inspire future research that could lead to agents with dynamic personalization behaviors. * The experimental results are clean and very clearly support the conclusions. * The relation to meta-learning is a nice connection that makes a lot of sense. Is the persective that this is an instance of meta-learning or merely related to meta-learning? Weaknesses --- There are two somewhat minor weakness: presentation and some missing related work. The main points in this paper can be understood with a bit of work, but there are lots of minor missing details and points of confusion. I've listed them roughly in order, with the most important first: * What factors varied in order to compute the error bars in figure 2? Were different random initializations used? Were different splits of the dataset used? How many samples do the error bars include? Do they indicate standard deviation or standard error? * L174: How exactly does the reactive baseline work? * L185: What does "without agent embeddings" mean precisely? * L201: More details about this metric are needed. I don't know exactly what is plotted on the y axis without reading the paper. Before looking into the details I'm not even sure whether higher or lower is good without looking into the details. (Does higher mean more information or does lower mean more information?) * Section 3: This would be much clearer if an example were used to illustrate the problem from the beginning of the section. * Will code be released? * L162: Since most experiments share perception between speaker and listener it would be much clearer to introduce this as a shared module and then present section 4.3 as a change to that norm. * L118: To what degree is this actually realized? * L84: It's not information content itself that will suffer, right? * L77: This is unnecessary and a bit distracting. * L144: Define M and N here. * L167: What is a "sqeuence of episodes" here? Are practice and evaluation the two types of this kind of sequence? Missing related work (seems very related, but does not negate this work's novelty): * Existing work has tried to model human minds, especially in robotics. It looks like [2] and [3] are good examples. The beginning of the related work in [1] has more references along these lines. This literature seems significantly different from what is proposed in this paper because the goals and settings are different. Only the high level motivation appears to be similar. Still, the literature seems significant enough (on brief inspection) to warrent a section in the related work. I'm not very familiar with this literature, so I'm not confident about how it relates to the current paper. [1]: Chandrasekaran, Arjun et al. “It Takes Two to Tango: Towards Theory of AI's Mind.” CVPR 2017 [2]: Butterfield, Jesse et al. “Modeling Aspects of Theory of Mind with Markov Random Fields.” International Journal of Social Robotics 1 (2009): 41-51. [3]: Warnier, Matthieu et al. “When the robot puts itself in your shoes. Managing and exploiting human and robot beliefs.” 2012 IEEE RO-MAN: The 21st IEEE International Symposium on Robot and Human Interactive Communication (2012): 948-954. Suggestions --- * L216: It would be interesting to realize this by having the speaker interact with humans since the listeners are analogous to role humans take in the high level motivation. That would be a great addition to this or future work. Final Justification --- Clarity - This work could significantly improve its presentation and add more detail, but it currently is clear enough to get the main idea. Quality - Despite the missing details, the experiments seem to be measuring the right things and support very clear conclusions. Novelty - Lots of work uses reference games with multiple agents, but I'm not aware of attempts to specifically measure and model other agents' minds. Significance - The work is a useful step toward agents with a theory of mind because it presents interesting research directions that didn't exist before. Overall, this is a pretty good paper and should be accepted. Post-rebuttal Updates --- After reading the reviews and the rebuttal this paper seems like a clear accept. After discussion with R3 I think we all roughly agree. The rebuttal addressed all my concerns except the minor one listed below satisfactorily. There is one piece R3 and I touched on which is still missing. I asked about the relation to meta-learning and there was no response. More importantly, R3 asked about a comparison to a practice-stage only reward, which would show the importance of the meta-learning aspect of the reward. This was also not addressed satisfactorily, so it's still hard to understand the role of practice/evaluation stages in this work. This would be nice to have, but rest of the paper provides a valuable contribution without it. Though it's hard to tell how presentation and related work will ultimately be addressed in the final version, the rebuttal goes in the right direction so I'll increase my score as indicated in the Improvements section of my initial review.

Reviewer 3



Highlighting strengths and weaknesses of the submission below. Strengths a. The paper is generally easy to follow. The authors do a decent job of motivating the problem / task at hand and the key axes of variation (different perception machinery and different degrees of conceptual understanding). b. Apart from the issues highlighted under weaknesses, the authors do a decent job of describing details associated with the experiments section. Within the controlled setup, the heuristics used to construct the population of listeners, modeling different degrees of understanding in listeners, etc. is explained clearly. Weaknesses a. The paper does not discuss some amount of recent related work in the space of reference games (using images or otherwise) and discrete communication (single / multiple time-step episodes), using dialog (natural language or otherwise) or single-round QA. Some pointers being https://www.aclweb.org/anthology/D17-1321, https://arxiv.org/pdf/1904.09067.pdf (deals with a population of agents) and https://arxiv.org/pdf/1703.06585.pdf. The authors highlight (L73 - 75) that the focus of the works on emergent language / communication is on the communication protocol unlike that of this work (which is on the agent’s mental model of other agents). It’s not entirely clear if the former is independent of the latter. This, combined with the fact that performance in these settings is mostly reported in context of the downstream performance, implies that the existing experimental settings could’ve been built on top of. Could the authors comment on this? Was there any other reason except for the fact that learning communication in conjunction to models of the listener adds another stochastic component? b. In section 3.3, under “Attribute Selection Policies”, it is mentioned that the speaker has access to a parameterized policy and a function V (s_k, a_k). It is unclear what the function V is trained on / how it is trained. The equation above L144 seems to suggest it is trained over the train (practice) and evaluation runs. But the following sentence implies evaluation sessions are handled greedily w.r.t. V. Finally, the structure of the cumulative reward function used to train the active policy is confusing. Why isn’t just the cumulative reward \sum_k r_k used as R? The provided reasoning is equally confusing -- “encouraging the … practice episodes” -- in the sense that an informative embedding (given perfect communication) should ideally form even if R = \sum_k r_k. Unless I am missing something, this choice seems odd. Can the authors comment on / clarify this? c. How do the authors sample the target and confounder images? My major concerns with the paper revolve around the ones mentioned above under weaknesses. The mentioned weaknesses also affect the clarity and originality of the paper to some extent. The paper would benefit from addressing the same. -------------------------------------- Post-Rebuttal Updates -------------------------------------- Thanks to the authors for responding to the concerns raised in the reviews. The rebuttal satisfactorily addressed my concerns regarding situating the work properly in context of recent related work in this sub-space. While the process described in L143-151 is easy to follow, I found the choice of training V(s_k, a_k) on practice + evaluation runs and choosing greedily w.r.t. the same (L144-145) odd at first as this was not stated in context of a continually adaptive speaker in this specific section. This was somewhat evident from how the rewards for the active policy were structured (above L148; error of V on practice episodes) but was not clear enough. R2’s comments and the author response helped alleviate this concern. I would suggest the authors to make this more clear in the final version. As R2 and I pointed out, the paper would benefit from discussing in detail a comparison to a practice-stage reward setting so as to highlight the importance of the meta-learning aspect of the approach. The comment that I had made in this context was regarding how the reward for training the active policy was structured and the associated justification provided in L148-150. I specifically asked how the current choice (error in V over evaluation episodes) compares to just maximizing the reward over practice episodes and whether that would / would not lead to informative agent embeddings. The response in the rebuttal document pointed to the fact that “epsilon greedy leads to more informative embeddings in spite of the similar downstream performance as the active policy” — which seemed orthogonal to the question I had asked. While I believe this does not negate the contributions of the paper but it does seem like an important point to clarify / address to justify the training paradigm. Having said all the above, I generally like the paper and the experimental analyses seem sound and thorough. I agree with R2 that the work is well-positioned to inspire interesting work in this setting. Therefore, I am increasing my score to 7.

[Author Response · NeurIPS 2019]

We sincerely thank all reviewers for their feedback. We present an image reference game where it is necessary to model
other agents' understanding of task-related concepts to succeed. The reviewers indicate that our framework is shown
through a technically sound experimental evaluation (R1) to be capable of modeling other agents' expertise (R2,R3),
has relevance to human interaction domains (R1,R2), and is well positioned to inspire future research in settings with
dynamic agent behaviors (R2). We will clarify the issues raised below and incorporate them into our final version.

**R3: Training V and active policy.** V is trained by minimizing MSE on all practice and evaluation games independent
of the speaker variant. Attributes in evaluation games are chosen greedily using $V$, in practice games different
parameterized policies are used. Active policy explicitly minimizes the MSE for $V$. Optimizing directly for high reward,
i.e. $R = \sum_k r_k$, does not necessarily increase information content for understanding the listener; e.g. active and epsilon
greedy policies obtain equivalent downstream performance (Fig2) but active policy achieves lower VI (Fig4).

**R2: Reactive baseline, define N + M (L144).** This is a static policy which always uses the same attribute until a
negative reward is encountered (L167), at which point a new attribute is sampled. "Sequence of episodes" refers to
all of the $N + M$ games played with each listener, respectively defined as the number of practice games the speaker
uses to understand the listener, and the number of evaluation games to verify the speaker's model. Reactive baseline
remembers utilized attributes when going from practice to evaluation games, disregarding attributes that didn't work.

**R1, R2: Qualitative example.** As an example, we train a speaker with 5 listener
populations that respectively do not understand color, shape, size, length, and
pattern attributes. Due to the lack of space, we show two games from a randomly
sampled test set sequence of the trained speaker policy with a color blind agent.
We will extend this with more examples in the main paper.

| Game | Maximally discrim. | Chosen | Target | Confounder | Why? |
|---|---|---|---|---|---|
| 6 | Green breast | Green breast | | | |
| 76 | Yellow leg | Upland ground like shape | | | Listener is colorblind |

**R2, R3: Related work.** Our work is indeed applicable to robotics: robots could
reason about users' understanding of object properties when describing object locations [C], as well as about multiple
other agents' intentions via a probabilistic generative model, which could extend our model to cooperative tasks [B].
Modeling users' understanding of an AI's mind [A] could provide an explanation component to teach users about the
AI. We would like to disentangle two orthogonal aspects of communication, i.e. modeling of other agents and language
learning. While [D,E,F] focus on language learning ([D,E] present synthetic reference games and [D,F] use two agents),
we focus on agent modeling on real-world images on a population of agents as they may have different capacities to
understand the task-related concepts. Our model with emergent language is an interesting extension.

**R2: Speaker's mental model of listener human-interpretable?** The clusters formed with agent embeddings (L194-
201) correlate well with the ground truth clustering of the population, since all speakers with agent embeddings gradually
minimize the VI score as more games are played (Fig4). Moreover, the performance increase with agent embeddings
in Fig3 suggests that the learned function $V$ encodes listeners' conceptual understanding. Since the attributes are
inherently interpretable, $V$'s output is a human-interpretable representation of the speaker's model of the listener.

**R2: Without agent embeddings.** After each game, the speaker incorporates the outcome into the agent embedding
(using an LSTM, Fig1), allowing it to form a model of the listener; the embedding is used to condition the attribute
selection policy, i.e. Description Generation Module. With "no agent embedding", a zero-vector carrying no information
between games is used. Hence, the speaker must maximize performance without storing information about the listener.

**R2: Error bars in Fig2, R3: sampling of target and confounder.** Each experiment was run with 3 seeds (random
initializations). Error bars represent the standard deviation. Dataset splits, i.e. training and test images, remain the same
across seeds. Target and confounder images are randomly sampled from the training set (during training) or test set (at
test time). Practice and evaluation games occur both during training (with parameter updates) and testing (no updates).

**R2: Variation of Information metric.** $VI(C, C') = H(C) + H(C') - 2I(C, C')$ measures how much information
is lost or gained by switching from a clustering $C$ to $C'$ where $H$ is the entropy of a clustering and $I$ is the mutual
information between two clusters. The lower VI the better, entailing a close correlation between clusters.

**R2: Shared perception module.** As in reality, confusion between different agents may occur also due to how they
perceive their environment. To maintain generality, we allow our framework to have separate perception modules.

**R2: Is the information content or the message lost? (L84)** The information content itself does not suffer due to the
listener's misunderstanding, since the speaker communicates losslessly. The listener's ability to use the information in
the message suffers, since it is difficult for the listener to properly compare images using a poorly understood attribute.

**R1,R2: Writing and Code.** We will remove "Ties to RL" (L77); release code, data and models upon acceptance.

**[A]** Chandrasekaran et al., It Takes Two to Tango: Towards Theory of AI's Mind, CVPR 2017 **[B]** Butterfield et al.,Modeling
Aspects of Theory of Mind with Markov Random Fields, IJSR 2009 **[C]** Warnier et al., When the robot puts itself in your shoes.
Managing and exploiting human and robot beliefs, IEEE RO-MAN 2012 **[D]** Kottur et al., Natural Language Does Not Emerge
'Naturally'in Multi-Agent Dialog, EMNLP 2017 **[E]** Cogswell et al., Emergence of Compositional Language with Deep Generational
Transmission, arXiv 2019 **[F]** Das, et al., Learning cooperative visual dialog agents with deep reinforcement learning, CVPR 2017


[Meta-Review · NeurIPS 2019]

Reviewers all voted to accept this submission and had their concerns generally addressed by the rebuttal. They were impressed by the clarity of the experimental setting and empirical results.